# The Wheat Intrinsically Disordered Protein *Td*RL1 Negatively Regulates the Type One Protein Phosphatase *Td*PP1

**DOI:** 10.3390/biom15050631

**Published:** 2025-04-28

**Authors:** Fatma Amor, Mariem Bradai, Ikram Zaidi, Vitor Amorim-Silva, Nabil Miled, Moez Hanin, Chantal Ebel

**Affiliations:** 1Functional Genomics and Plant Physiology Laboratory, Institute of Biotechnology, University of Sfax, P.O. Box 1175, Sfax 3038, Tunisia; 2Biotechnology and Plant Improvement Laboratory, Centre of Biotechnology of Sfax, P.O. Box 1177, Road Sidi Mansour km 6, Sfax 3018, Tunisia; 3Área de Mejora y Fisiología de Plantas, Instituto de Hortofruticultura Subtropical y Mediterránea “La Mayora”, Universidad de Málaga-Consejo Superior de Investigaciones Científicas (IHSM-UMA-CSIC), Universidad de Málaga, 29010 Málaga, Spain; 4Department of Biological Sciences, College of Science, University of Jeddah, Jeddah 23445, Saudi Arabia

**Keywords:** PP1, *Td*RL1, phosphatase activity, PP1-interacting proteins, three-dimensional structure prediction

## Abstract

Type 1 protein phosphatases (PP1s) are crucial in various plant cellular processes. Their function is controlled by regulators known as PP1-interacting proteins (PIPs), often intrinsically disordered, such as Inhibitor 2 (I2), conserved across kingdoms. The durum wheat *Td*RL1 acts as a positive regulator of plant stress tolerance, presumably by inhibiting PP1 activity. In this work, co-immunoprecipitation and bimolecular fluorescence complementation (BiFC) assays demonstrate that the durum wheat *Td*PP1 interacts with both *Td*RL1 and *At*-I2 in vivo. YFP fluorescence restored after *Td*RL1-*Td*PP1 interaction decorated specifically the microtubular network of the tobacco co-infiltrated cells. In vitro phosphatase assays revealed that *Td*RL1 inhibited the activity of wild-type *Td*PP1 and two mutant forms (T243M and H135A) in a concentration-dependent manner, showing a novel and potent inhibition mechanism. Structural modeling of the *Td*PP1-inhibitor complexes suggested that both *At*-I2 and *Td*RL1 bind to *Td*PP1 by wrapping their flexible C-terminal tails around it, blocking access to the active site. Remarkably, the model showed that *Td*RL1 differs from *At*-I2 in its interaction with *Td*PP1 by trapping the phosphatase with its N-terminal tail. These findings provide important insights into the regulatory mechanisms governing the activity of PP1s in plants and highlight the potential for targeted inhibition to modulate plant stress responses.

## 1. Introduction

Type 1 protein phosphatases (PP1s) are ubiquitous serine/threonine protein phosphatases that play essential roles, across eukaryotes, in various cellular processes including cell cycle progression and meiosis, protein synthesis, as well as the regulation of glucose metabolism, transcription, and cytoskeleton organization [1,2].

In the plant kingdom, PP1s exhibit remarkable functional diversity, as they have been shown to be engaged in many pathways including but not limited to hormone signaling [3], light responses [4], and cell wall integrity [5]. For instance, TOPP4, one of the nine *Arabidopsis thaliana* PP1s, is a key regulator of the gibberellic acid (GA) signaling pathway [6]. By promoting the degradation of DELLA proteins (through their dephosphorylation), the negative regulators of GA responses, TOPP4 allows GA to stimulate plant growth and development [6]. TOPP4 interacts also with the PP1 Regulatory Subunit 3 (PP1R3), which is essential for regulating its phosphatase activity and nuclear localization. In *pp1r3* mutant plants, TOPP4 activity is dysregulated, resulting in abscisic acid (ABA), salt, and drought hypersensitivity [3]. The multiple roles of TOPP4 have been revealed hence from the T247M dominant-negative mutation [6], which results in pleiotropic phenotypic defects [7]. Furthermore, ATUNIS1 (AUN1), also known as TOPP9, along with its homolog ATUNIS2 (AUN2, TOPP8), play a critical role in maintaining cell wall integrity in tip-growing plant cells, such as pollen tubes and root hairs. TOPP9 function in cell wall formation was revealed in a screen for suppressors of the *anx1 anx2* mutations encoding CrRLK1L receptor-like kinases in which *aun1* harbors a D94N mutation affecting the conserved Asp of the catalytic site, leading to a dominant-negative phospho-dead protein [5].

*Td*PP1, the only characterized PP1 in durum wheat so far, has been reported to be involved in the control of Brassinosteroid (BR) signaling by binding to and dephosphorylating the BR-INSENSITIVE-EMS-SUPPRESSOR1 (BES1) transcription factor [8]. Indeed, the overexpression of *Td*PP1 in *Arabidopsis* leads to BR hyposensitivity illustrated by increased primary root length in response to external BR application. Moreover, these lines exhibit an increased auxin-mediated root plasticity upon exposure to salt or osmotic stress [9]. Likewise, transgenic rice lines overexpressing *Os*PP1a confer improved tolerance to salt stress, with increased expression of stress-responsive genes involved in antioxidant defense [10]. This underlines the crucial regulatory role of PP1s in modulating plant salt stress tolerance, probably via the control of hormonal pathways. It is well established in different eukaryotes that active PP1 holoenzymes typically exist as heterodimers composed of a well-conserved catalytic subunit (PP1c) and a structural or regulatory subunit known as a PP1-interacting protein (PIP) [1,11,12]. PP1c consists of a single-domain protein of 35–38 kDa, characterized by an α/β fold and a central β-sandwich structure featuring a channel for catalytic activity, where manganese (Mn^2+^) and iron (Fe^2+^) bind to the highly conserved amino acids, aspartate and histidine within the catalytic DxVDRG motif and the GNHE motifs, respectively [13,14,15]. Asp is critical for coordinating metal ions in the active site, while His is involved in proton transfer during the dephosphorylation reaction [14,15,16]. The versatile roles of PP1s are dictated by the binding of various regulatory proteins (or PIPs) that control substrate specificity and subcellular localization, allowing functional pleiotropy. From studies in mammals, 90% of the PIPs are intrinsically disordered proteins (IDPs), allowing both flexible binding and the formation of stable complexes with PP1. Seventy percent of the 200 PIPs identified share a common binding motif known as RVxF (K/RK/RV/IxF/W) [1,11], which allows binding to a hydrophobic groove on PP1 away from the active site [17,18]. In nearly 10% of RVxF-containing PIPs, one can find the SILK motif (G/SILR/K), located upstream of RVxF motif, which helps the binding of PIPs to the catalytic PP1 [19]. The MyPhoNE motif (RxxQV/I/LK/RxY/W) found in some PIPs, such as MYPT1, contributes to PP1 binding through a five-turn α-helix that interacts with residues (R96, H125, Y134, R221, Y272) located in a hydrophobic cleft on the surface of PP1 [19].

Inhibitor-2 (I2) is recognized as one of the most ancient and conserved regulators of PP1s [20]. I2 is a small IDP that plays a dual role as a loader of active-site metals and a competitive inhibitor of PP1 depending on its phosphorylation status or acts as a molecular chaperone to promote proper PP1 folding [20]. I2 binds to PP1 through several conserved motifs, including the canonical RVxF and non-canonical SILK motif [20]. The *Arabidopsis* I2 inhibitor (*At*-I2), which harbors an RVxF motif, is able to inhibit all nine *Arabidopsis* PP1 isoforms [21] and has been shown together with TOPP1 to negatively regulate the ABA signaling pathway by inhibiting the SNF1-related protein kinase 2 (SnRK2) [22]. Genetic screens performed by Ogawa et al. [23] allowed the identification of Rice Salt Sensitive 1 (RSS1) protein, an IDP that binds to *Os*PP1a [24] and probably inhibits its activity, contributing to meristem maintenance in rice under salt stress by promoting cell cycle entry. The durum wheat *Td*RL1 protein, the structural and functional ortholog of RSS1, is also an IDP that complements the *rss1* salt-induced dwarfism and promotes stress tolerance in rice and *Arabidopsis,* hence suggesting that *Td*RL1 might also regulate PP1 to confer stress tolerance [25,26].

Here, we demonstrate, by co-immunoprecipitation and BiFC assays, the molecular interaction between the two durum wheat proteins *Td*PP1 and *Td*RL1. In vitro assays showed that *Td*RL1 inhibits the phosphatase activity of *Td*PP1. Structural modeling of the interactions between *Td*PP1 and its regulators, *Td*RL1 and the *Arabidopsis* I2 inhibitor, is also reported. Our data provide evidence that *Td*RL1 acts as a PIP that negatively regulates the *Td*PP1’s phosphatase activity by blocking access to the catalytic site.

## 2. Materials and Methods

### 2.1. Nicotiana benthamiana Growth Conditions and Agro-Infiltration Assays

*Nicotiana benthamiana* seeds were sown directly on soil and the resulting seedlings were grown for 4 weeks at 22 °C with 70% relative humidity with a photoperiod of 16 h light/8 h dark. *Agrobacterium*-mediated transient expression in tobacco leaf epidermal cells was then performed as described in Bradai et al. [8]. Briefly, *Agrobacterium tumefaciens* strain *GV3101* was transformed with the different constructs described hereafter and grown in liquid LB containing the appropriate antibiotics overnight at 28 °C with shaking at 200 rpm. Cells were harvested by centrifugation (15 min at 3000 rpm at room temperature) and resuspended in an infiltration medium (10 mM MES pH 5.6, 10 mM MgCl_2_, 200 μM acetosyringone) to an OD_600_ of 1. For co-expression experiments, the *Agrobacterium* strains containing different constructs are mixed in a 1:1 ratio just before infiltration.

### 2.2. Molecular Cloning

*TdPP1* (acc. number KM203893) open reading frame was cloned in pCR2.1 and in pB7FWG2 [27] for a Ct GFP fusion as described by [8,28]. The *Td*PP1-H135A mutant form was obtained by performing two parallel PCR amplifications using Pfu polymerase (Bio Basic, Markham, ON, Canada). First, the 5′-end (P1 product, 550 bp) was amplified using *Td*PP1F3 (5′-TTAGGATCCATGGCGGCGGCGCCGGCGGC-3′) and P1mPhosR (5′-GAAGCACACTCAGCGTTGCC-3′) primers and the following program: 2 min at 95 °C, followed by 30 cycles of 95 °C for 30 s, 60 °C for 30 s, and 72 °C for 40 s. Secondly the 3′-end (P2 product, 400 bp) was amplified with PP1mPhosF (5′-GAGAGGCAACGCTGAGTGTGCTTC-3′) and *Td*PP1R3 (5′-TTAGTCGACTCACATCTTGTTTGACGCCA-3′) with the same thermal cycling conditions. A final PCR amplification was performed using *Td*PP1F3 (5′- TTAGGATCCATGGCGGCGGCGCCGGCGGC-3′) and *Td*PP1R3 (5′-TTAGTCGACTCACATCTTGTTTGACGCCA-3′) to amplify the complete open reading frame containing the mutated site using both P1 and P2 PCR products as templates with the following thermal cycling conditions: 2 min at 95 °C, then 30 cycles of 95 °C 30 s, 60 °C 30 s, and 72 °C 1 min. The resulting PCR product was cloned in *BamH*I and *Sal*I cloning sites of pET28a for an N-terminal fusion with the 6xHis tag. The presence of the H135A mutation was confirmed by sequencing.

The *Td*PP1-T243M mutation was generated using the QuickChange Lightning Site-Directed Mutagenesis Kit (Agilent Technologies, Santa Clara, CA, USA). A PCR amplification was performed using pDONRzeo-*Td*PP1 as template and the primer pair designed to change the original codon ACG to ATG (tdpp1 Fw (5′-GGTCATGACAAAGTTAtGGAGTTCCTTCTAAAG-3′), tdpp1 Rv (5′-CTTTAGAAGGAACTCCaTAACTTTGTCATGACC-3′)). The thermal cycling conditions were the following: an initial denaturation at 95 °C for 2 min, followed by 18 cycles of 95 °C for 20 s, 60 °C for 10 s, and 68 °C for 2 min 30 s, with a final extension at 68 °C for 5 min. Following amplification, *Dpn*I digestion was performed to remove the parental plasmid DNA, and the resulting plasmid was transformed into XL10-Gold ultracompetent cells according to the manufacturer’s instructions. The presence of the mutation was confirmed by DNA sequencing, and the *Td*PP1-T243M mutant was subsequently cloned into the *BamH*I and *Sal*I sites of pET28a, after a final amplification with the same primers used for the *Td*PP1-H135A mutation.

*TdRL1* (acc. number KJ944286) in pDON207 [26] was mobilized into pGWB14 [29] for a Ct fusion with HA tag by LR recombinase (Invitrogen, Waltham, MA, USA) as recommended. *Td*RL1 was subcloned into pDONRzeo by performing a BP reaction after linearization of pGWB14-*Td*RL1 using the *Nhe*I restriction enzyme. For its expression in *E. coli*, the pHMGW-*Td*RL1 construct (with MBP and His tags) described by [26] was used. The *At*-I2 open reading frame was cloned into pDON201 (kind gift of Prof. G. Moorehead, University of Calgary, Canada) and mobilized into pDEST15 and into pGWB14 [29] for HA fusion by an LR reaction resulting in pDEST-*At*-I2 (GST-tag) for the expression in *E. coli BL21* (DE3) and to pGWB-I2 for agro-infiltration, respectively.

pDONZeo-*Td*RL1 and pDON201-*At*-I2 were mobilized into pDEST-GW-VYNE for an N-terminal fusion (*At*-I2-nYFP; *Td*RL1-nYFP) by LR recombinase (Invitrogen) as recommended. BIK1-nYFP; BRI1-nYFP in pDEST-GW-VYNE were used a negative control, and pDEST-GW-VYCE (*Td*PP1-cYFP; BRI1-cYFP) was used for a C-terminal fusion with YFP [8,30]

### 2.3. Co-Immunoprecipitation Assays

Co-immunoprecipitation was performed as previously reported [31]. In brief, total proteins were extracted from 100 mg of tobacco agro-infiltrated leaves (with pGWB-*Td*RL1 or pGWB-I2 and pB7FWG2-*Td*PP1) in an extraction buffer (2 mL/g of powder in 50 mM Tris-HCl, pH 7.5; 150 mM NaCl; 10% glycerol; 10 mM EDTA, pH 8; 1 mM NaF; 1 mM Na_2_MoO_4_·2H_2_O; 10 mM DTT; 0.5 mM PMSF; 1% (*v*/*v*) P9599 protease inhibitor cocktail (Sigma-Aldrich, Saint-Louis, MO, USA); 0.5% (*v*/*v*) Nonidet P-40 (USBiological, Salem, MA, USAence)) at 4 °C for 30 min on constant end-over-end rocking. After a centrifugation (9000 rpm) for 20 min at 4 °C, supernatants were filtered by gravity through Poly-Prep Chromatography Columns (Bio-Rad Laboratories, Hercules, CA, USA) and 100 µL was saved as input for Western blot analysis. The remaining supernatants were incubated 2 h at 4 °C with 15 µL GFP-Trap coupled to agarose beads (Chromotek, Planegg, Germany). During the incubation of protein samples with GFP-Trap beads, the final concentration of Nonidet P-40 was adjusted to 0.2% (*v*/*v*) in all cases to avoid unspecific binding to the matrix as recommended by the manufacturer. Following incubation, the beads were collected and washed four times with a wash buffer (extraction buffer without detergent). Finally, beads were resuspended in 75 µL of 2x Laemmli Sample Buffer and heated at 60 °C for 30 min to dissociate immunocomplexes from the beads. Total (input), immunoprecipitated (IP), and co-immunoprecipitated (CoIP) proteins were separated by electrophoresis in 10% SDS-PAGE, analyzed by western blot analysis using anti-HA (1:3000, Sigma-Aldrich, Saint-Louis, MO, USA) or anti-GFP (1:600; Santa Cruz Biotechnology, Dallas, TX, USA) antibodies. Appropriate peroxidase-conjugated secondary antibodies were added as follows: anti-mouse IgG whole molecule peroxidase (1:80,000; Sigma-Aldrich, Saint-Louis, MO, USA) for anti-GFP, and anti-rabbit IgG whole molecule peroxidase (1:80,000; Sigma-Aldrich, Saint-Louis, MO, USA) for anti-HA.

### 2.4. Bimolecular Fluorescence Complementation (BiFC)

The different DNA constructs were used to agro-infiltrate *N. benthamiana* leaves as described above. Epidermal leaf cells were observed by confocal microscopy (Leica TCS SP5 II, Leica Microsystems, Wetzlar, Germany) two days after co-infiltration. The settings used for the laser scanning are 405 nm excitation and detection at 425–475 nm for calcofluor-white and 514 nm excitation and detection at 525–570 nm for YFP.

### 2.5. Production and Purification of Recombinant Proteins

The recombinant proteins (His::*Td*PP1 forms, GST::*At*-I2, and His-MBP::*Td*RL1) were produced after transforming *E. coli BL21* (DE3) competent cells with the corresponding constructs. For each construct, 5 mL LB supplemented with the appropriate antibiotics was inoculated with a single colony cultured overnight at 37 °C at 200 rpm. The next day, 250 mL of LB supplemented with the corresponding antibiotics was inoculated with 2.5 mL of the starting culture and incubated at 37 °C until an OD of 0.5–0.6 was reached. The production was induced with 1 mM IPTG for 4 h at 37 °C. Then, after harvesting the cells by centrifugation at 6000 rpm for 20 min at 4 °C, the cell pellet was resuspended in 50 mL lysis buffer (50 mM Tris-HCl pH 7.5, 100 mM NaCl, 0.5% Triton X-100, 1 mM DTT, 1 mM PMSF, 1 mg/mL lysozyme) and incubated for 30 min at 37 °C with gentle shaking. After centrifugation for 30 min at 4 °C at 10,000 rpm, cell debris were removed, and the supernatant was collected. For *Td*RL1 and *Td*PP1 that carry a 6xHis tag, purification was performed on a nickel column (Takara, Kyoto, Japan) as recommended by the supplier with an elution buffer (50 mM Tris-HCl pH 7.5, 100 mM NaCl, 500 mM imidazole). For the GST-tagged *At*-I2 protein, cell lysis was performed using 0.5% Triton X-100, 1 mM DTT, 1 mM PMSF, and PBS 1x, 1 mg/mL lysozyme. After incubation for 30 min at 37 °C and centrifugation, the obtained supernatant was passed through a column containing glutathione Sepharose 4B resin (Cytiva, Marlborough, MA, USA), previously washed with PBS 1x buffer. Elution was carried out with 10 mM reduced glutathione (Bio Basic Inc., Markham, ON, Canada) at pH 8. Concentrations of the recombinant proteins were determined by the Bradford Assay [32] as recommended by the manufacturer (Sigma-Aldrich, Saint-Louis, MO, USA).

### 2.6. Phosphatase Assays

The activity of *Td*PP1 protein was measured using p-nitrophenyl phosphate (*p*NPP; Bio Basic [21,33]) as a substrate. The reaction mixture (1 mL) contained the reaction buffer (50 mM Tris-HCl, pH 7.5, 150 mM NaCl, 1 mM MnCl_2_), 1 μg of PP1 enzymes (WT or mutants forms), and 5 mM of *p*NPP. The mixture was then incubated at 37 °C for 30 min. Measurement of phosphatase activity was performed by measuring the absorbance at 405 nm (Jenway™ 72 Series UV/Visible Spectrophotometer, Jenway, Stone, UK). To test their effect on PP1 phosphatase activity, increasing concentrations of *At*-I2 or *Td*RL1 were incubated with the enzyme for 10 min at 37 °C before adding the reaction mixture. The experiment was repeated at least 3 times with independently produced proteins and each experiment comprised at least three technical replicates. The specific activity of PP1 was calculated using the molar extinction coefficient ε = 18,000 M^−1^ cm^−1^, corresponding to *p*NPP and plotted as a function of the inhibitor concentration in nM. The IC50 values were calculated with the following method: the inhibition curve was generated based on the percentage of inhibition at three different inhibitor concentrations. The data were analyzed using the linear equation of the curve, Y = MX + C, where Y represents the percentage of inhibition, M is the slope, and C is the intercept. The IC50 value (X) was determined by setting Y to 50% inhibition and resolving the equation X = (50-C)/M.

### 2.7. Statistical Analysis

The statistical analyses were performed using One-Way Analysis of Variance (ANOVA) followed by Tukey’s post hoc test. This allowed for comparison of the inhibitory effects of the *At*-I2 and *Td*RL1 inhibitors across multiple groups. The IC50 values, representing the half-maximal inhibitory concentration, were calculated for each inhibitor and expressed as the mean ± standard deviation.

### 2.8. Homology Modeling and Complex Structure Optimization

*Td*PP1 wild-type, H135A, and T243M mutants were modeled using the automated modeling server “swiss model” [34] based on the structure of the mouse phosphatase in complex with mouse I2 (pdb code 2O8A) [35]. The full length *At*-I2 model was extracted from AlphaFold models (Q9LTK0.1.A, Protein phosphatase inhibitor 2 [36]). The fragment 110–137 of mouse I2 was the main interacting region with the phosphatase active site. Corresponding interaction residues in *Td*PP1 were used to orient the generation of the complex between the phosphatase and the full length *At*-I2 by Zdock server (Zdock v3.0.2) [37]. Contact residues selected from *Td*PP1 were R106, I140, R142, I143, Y144, D207, D230, R231, V260, E262, E266, Y282, C283, E285, and F286. The energy of the complex was then optimized using three steps of conjugate gradient, each of 40 cycles. The models of the *Td*PP1-H135A and *Td*PP1-T243M mutants in complex with the *At*-I2 were generated using the same strategy described for the wild-type protein.

The model of *Td*RL1 was extracted from the AlphaFold database (A0A3B6PNT1.1.A, Uncharacterized protein [36]). Contact residues R106, I140, R142, I143, Y144, D207, D230, R231, V260, E262, E266, Y282, C283, E285, and F286 from *Td*PP1 were defined as interacting residues and the complex structure between the *Td*PP1; the *Td*PP1-H135A or the *Td*PP1-T243M mutants with the *Td*RL1 inhibitor were then generated by ZDock. The energy of each complex was then optimized using three steps of conjugate gradient, each of 40 cycles, implemented to the swisspdb viewer program [38]. Structure visualization, interaction analysis, and figure generation were carried out using the Biovia Studio visualizer (Discovery Studio Modeling Environment, Release 2017 R2, 2007; Accelrys Software Inc., San Diego, CA, USA).

## 3. Results

### 3.1. TdRL1 Interacts with TdPP1

The fact that *Td*RL1 is functionally orthologous to RSS1 and promotes stress tolerance suggests that the RSS1-PP1 signaling pathway is conserved in rice and wheat [25,26,39]. To better understand how the *Td*RL1-PP1 pathway operates, we investigated here the interaction between *Td*PP1 and *Td*RL1 and its effect on the *Td*PP1′s catalytic activity. For this purpose, we first performed co-immunoprecipitation experiments using anti-GFP beads to immunoprecipitate *Td*PP1::GFP and an HA-tagged *Td*RL1 after co-infiltration of *N. benthamiana* leaves with Agrobacterium strains harboring the corresponding constructs. As shown in Figure 1, the immunoblot revealed a clear interaction between *Td*PP1 and *At*-I2, which was used as a positive control [28]. This interaction is illustrated by a prominent band for *At*-I2 in the *Td*PP1-GFP IP lane. More importantly, *Td*RL1::HA co-immunoprecipitated with *Td*PP1::GFP and this interaction is specific, since the band was absent in the IP when the tobacco leaves were co-infiltrated with a construct harboring GFP alone as negative control. These results indicate that *Td*RL1, like I2, interacts with *Td*PP1.

In addition, we performed Bimolecular Fluorescent Complementation (BiFC) assays to further investigate the interaction between TdPP1 and *TdRL1* in vivo and to explore the sub-cellular localization of the *TdRL1*-TdPP1 complex. Our results showed that TdPP1 interacts with *At*-I2 (Figure 2C). The restored YFP-fluorescence triggered by *At*-I2 and TdPP1 binding was observed both in the cytoplasm and in the nucleus. The co-infiltration with TdPP1-cYFP and TdRL1-nYFP also allowed complementation and hence emission of YFP fluorescence in tobacco cells. Most interestingly, the fluorescent signal lights up the cortical microtubular network, indicating that the TdRL1-PP1 complex might play a role within microtubular network. Both interactions are specific, since the YFP fluorescence restoration was not observed when *At*-I2 or *Td*RL1-nYFP were co-infiltrated with BRI1-cYFP used as a control (see Figure 2 and Appendix A) that interacts with BIK1 (BOTRYTIS-INDUCED KINASE 1), as previously shown [40]. Therefore, both co-IP and BiFC assays show that *Td*PP1 interacts with *Td*RL1 and *At*-I2 with differential cellular localization for both complexes.

### 3.2. TdPP1’s Activity Is Inhibited by TdRL1

Since PP1s are catalytic holoenzymes, the activity of which is directed by specific regulators [41], we investigated whether the binding of TdRL1 influences the phosphatase activity of TdPP1 in vitro using pNPP as a substrate. To this end, recombinant forms of TdPP1 (6xHis::TdPP1a) and TdRL1 (6xHis-MBP::TdRL1) were produced in E.coli and purified (Appendix A). We pre-incubated TdPP1 with increasing concentrations of TdRL1 for 10 min prior to the addition of the pNPP substrate and phosphatase activity measurements. We confirmed that TdPP1 is an active phosphatase [28], and the results clearly demonstrate that TdRL1 inhibits TdPP1 activity (Figure 3A). Indeed, with a concentration of 26 nM corresponding to a 1:1 molar ratio, TdRL1 inhibits the activity of TdPP1 by 59%. This inhibitory effect became more pronounced at higher concentration of TdRL1 reaching 89.7%. The purified GST::At-I2, a well-known negative regulator of PP1s, inhibited TdPP1’s activity more strongly than TdRL1, with a maximum inhibition of 90.4% for the highest concentration used. These findings strongly suggest that TdRL1 is a negative regulator of TdPP1.

To further explore how changes in TdPP1 might affect the interaction with regulators, mutated forms of TdPP1 were generated. Firstly, a mutant form where Thr243 in TdPP1 was replaced by a Met residue in a way to mimic the dominant negative mutant topp4-1 of Arabidopsis was used [6]. Indeed, this mutant harbors an EMS-induced missense mutation of Thr247 to methionine [6] and exhibits pleiotropic phenotypic defects [7]. It is noteworthy that neither Thr243 in TdPP1 nor Thr247 in TOPP4 are conserved within PP1s but are putative phosphorylation sites (Appendix A). Hence the modification of a phosphorylable polar residue to a hydrophobic residue might affect the activity and the binding of regulators. Alongside this, we also mutated the His135 residue of the conserved catalytic GNHE site (Appendix A) to alanine, which was expected to abolish the catalytic activity as it was previously reported on the phosphatase-dead form of Arabidopsis AUN1 protein [5]. In the absence of regulators, the H135A isoform exhibited only 16.44% residual activity relative to the WT, highlighting the critical role of the histidine residue in the TdPP1 function (Figure 3A). Interestingly, the activity of the T243M isoform retained 71.01% of WT PP1 activity (Figure 3A), indicating that this mutation decreased also the PP1 potential to dephosphorylate its substrate.

The inhibition potential of At-I2 on the activities of both mutant forms was then investigated (Figure 3A). The activity of TdPP1-T243M in the presence of At-I2 was reduced in a similar proportion to the inhibition observed with the wild-type enzyme, suggesting that the T243M mutation did not significantly impact the regulation potential of At-I2. Notably, the TdPP1 H135A mutant exhibited the highest level of inhibition by At-I2, with inhibition levels reaching 100% at high concentration. This inhibition was proportional to the activity of this mutated form, which was the weakest.

Both PP1 mutants, *Td*PP1-T243M and *Td*PP1-H135A, were slightly less sensitive to inhibition by *Td*RL1 compared to the wild-type PP1. The *Td*PP1-T243M mutant showed a lower inhibition level, suggesting that this mutation may decrease the enzyme’s sensitivity to the inhibition by *Td*RL1. The activity of the *Td*PP1-H135A mutant form at high concentrations was inhibited by approximately 91% by *Td*RL1 and might indicate that the *Td*PP1 residues, T243 and H135, may participate in the interaction with *Td*RL1 (Figure 3A).

The IC50 values (Figure 3B) also confirmed that *Td*RL1 and *At*-I2 have significant effects on PP1 activity. The IC50 of *Td*PP1 in the presence of *Td*RL1 was 32 ± 2.9 nM, while the IC50 values of *Td*PP1-T243M and *Td*PP1-H135A were 35.6 ± 1.98 nM and 37 ± 3.7 nM, respectively. In contrast, *At*-I2 showed more effective inhibition, with lower and similar IC50s for the different forms of PP1 varying from 26.7 ± 2.8 nM to 28.4 ± 1.5 nM.

### 3.3. Structural Study of At-I2 and TdRL1 Interactions with TdPP1

To underpin the molecular basis of the *Td*PP1 interactions with either *At*-I2 or *Td*RL1, we performed structural modeling using the well-known structures of mammalian PP1s. Our analysis shows that the phosphatase domain of *Td*PP1 was well superimposed to that of the mouse enzyme used as template (pdb code 2O8A) (Figure 4A,B). The root mean squared deviation (RMSD) between the two structures was 0.1 Å. Most of the mouse I2 structure was disordered and only some peptides were observed in the crystal structure of the template. The mouse I2 structure extracted from the template structure (residues 132–164) was superimposable to that of the *At*-I2 110–137 fragment (Figure 4B) since both peptides shared 30% amino acid identity. This suggests similar interactions with the phosphatase active site. The main interactions stabilizing *Td*PP1 and its inhibitor I2 in the template structure involved residues corresponding to R106, H135, S139, R142, I143, Y144, V260, E262, and F286 of *Td*PP1. These interactions were used to orient the model of the complex *Td*PP1/*At*-I2 using Zdock server (Zdock v3.0.2). The best poses generated by Zdock showed the inhibitor enrolling its C-terminal tail around the enzyme (Figure 4A). The long-disordered C-terminal part of the *At*-I2 inhibitor winds around the phosphatase structure (Figure 4A,B). The complex corresponding to the highest binding score showed that interactions between *At*-I2 and *Td*PP1 involved several segments of the inhibitor. The helix-turn-helix, corresponding to residues 90–146, seems to block the active site entry, explaining the inhibitory power of I2 towards the phosphatase (Figure 4B). This fact was supported by the superimposition of the *At*-I2 132–164 residues with the mouse 110–137 I2 fragment (Figure 4B). The interactions between the enzyme and I2 were mainly electrostatic. The complex was stabilized by ion pairing between R106, K160, D207, and R271 from the phosphatase and E128, E145, E144, and E13 from the inhibitor, respectively. Other electrostatic interactions involved K151 and K221 from the phosphatase and E145 and D106 from I2, respectively. Hydrophobic and hydrogen bonding interactions also contributed to stabilize the complex. These H-bonds were established between residues Y124, V156(N), R271, T298, L299(O), R271, and Y235 from the enzyme and residues E128, E145, W11(O), R6, G7(N), V9(N), and S92 from the inhibitor, respectively.

The generation of the models of the mutants H135A and T243M in complex with *At*-I2 yielded similar complexes to the wild-type. This was foreseen due to the high number of residues taking part in the interactions minimizing the effect of a single mutation on the complex configuration. The catalytic histidine H135 interacted with R106, which was stabilized through ionic interaction with E128 of the inhibitor. A mutation of the catalytic histidine to alanine would impair the activity but also weaken the interactions with the inhibitor. This might explain the reduction in the inhibition extent of H135A mutant by *At*-I2 at a one-to-one molar ratio, as compared to the wild-type enzyme. The N-terminal tail of *At*-I2 was located close to T243 of *Td*PP1 (Figure 4A,B) and stabilized through hydrophobic interactions at the surface of the phosphatase. T243 is not a main interacting residue with the inhibitor. This explains that the inhibition potential of *At*-I2 was not lost on the T243M mutant. Nevertheless, T243 was located at the enzyme surface, close to the inhibitor binding site (Figure 4B). Its replacement with M would slightly disturb these interactions due to the replacement of a polar residue by the bulky and hydrophobic methionine residue. This might explain a decrease and not a loss of the inhibitory power of *At*-I2 upon T243M mutation.

Similar to the observation of the phosphatase-I2 interactions, the predictions of the *Td*PP1-*Td*RL1 complex by Zdock (Figure 5A,B) showed *Td*RL1 holding the phosphatase tightly like a thread around a spinning top. The predictions of the complex between *Td*PP1 and *Td*RL1 corresponded to the same overall interaction while the positioning of the tether tail around the phosphatase changed (Figure 5A). The complex with the highest affinity showed that the N-terminal helices were tethering the clip around the enzyme (Figure 5B,C) mainly due to ionic interaction E13-R129. Hydrophobic interactions involving *Td*RL1 residues L43, V46, P55, I57, V68, L69, L75, Q78, L86, A96, and W103 were also participating in this tethering. Interestingly, the R129-P154 helical fragment containing an RVxF motif (_151_PLLF_154_) engages in direct interactions with the enzyme’s active site (Figure 5B–D). The peptide blocks the entry of the catalytic pocket, which explains the inhibitory power of *Td*RL1 towards *Td*PP1. Interactions stabilizing the helical fragment are mainly hydrophobic involving residues Y282, F286, and Y144 from the phosphatase and L153 and W144 from *Td*RL1. Other hydrophobic interactions are established by the RVxF residues (_151_PLLF_154_) with Q259, V260, and V261 phosphatase residues located at the entry of the catalytic pocket. The interaction is strengthened by hydrogen bonding between E285 and R106 from *Td*PP1 and S145 and L141(O) from *Td*RL1, respectively. Furthermore, the C-terminal peptide of *Td*RL1 containing a second RVxF motif (_211_KPVEF_215_) is wrapped around the phosphatase catalytic site (Figure 5B,C) and stabilized through hydrophobic interactions at the enzyme’s surface. The F215 residue at the _211_KPVEF_215_ motif is stabilized by P206 from *Td*PP1. Interestingly, when superimposed to the phosphatase-*Td*RL1 structure complex, the mouse I2 inhibitor in complex with the phosphatase showed interactions with a similar region corresponding to the catalytic cavity entry (Figure 5D).

Phosphatase residue T243 is exposed at the enzyme surface, close to the active site and therefore, to the *Td*RL1 binding region (Figure 5C–E), without being a key component in the interaction. This explains the fact that its mutation by a bulky residue such as methionine slightly reduces the *Td*RL1 inhibitory power without impairing it. This might be due to the protein-inhibitor packaging where the inhibitor binds tightly to the enzyme, rendering the interaction less effective. The H135A mutant activity was drastically reduced due to the crucial role of catalytic histidine in the reaction. Catalytic histidine belongs to the interaction surface with the inhibitor, at the catalytic cavity entry. The mutation of H135 by alanine would weaken the interaction explaining a reduced inhibitory power on this “phosphatase-dead”-like mutant. Even though the complex of each mutant with *Td*RL1 did not show a drastic difference as compared to the wild-type enzyme, the mutations would slightly weaken the interactions and therefore the affinity of the inhibitor to the phosphatase, which would explain the lower inhibition rates recorded for the phosphatase mutants as compared to the wild-type enzyme.

In summary, *At*-I2 and *Td*RL1 share some common features of phosphatase inactivation. Both *At*-I2 and *Td*RL1 block the entry of the active site thus inhibiting the enzyme (Figure 5E). Both inhibitors are winding around the enzyme by an unstructured flexible tail. *At*-I2 and *Td*RL1 seem to inactivate the phosphatase in a similar way to the mouse I2, through blockage of the active site entry (Figure 5E).

## 4. Discussion

The multiple functions ensured by plant PP1s are dictated by wide range of PIPs, which are often structurally disordered. Here, we show that the wheat type one protein phosphatase *Td*PP1 and the intrinsically disordered protein *Td*RL1 are engaged in a physical interaction leading to PP1 inhibition that appears to be important for the regulation of plant abiotic stress responses. This finding is supported by a combination of co-immunoprecipitation and BiFC assays.

The BiFC assays showed the peculiarity of the *Td*PP1-*Td*RL1 complex with the reconstituted YFP fluorescence decorating the cortical microtubule network. This observation corroborates the previous data reported by Mahjoubi et al., evoking a potential involvement of *Td*RL1 in the microtubule network regulation [26]. Indeed, comparative co-expression data analysis between *Td*RL1 and its rice ortholog RSS1 revealed that common genes related to microtubule networks were overrepresented in co-expressed genes. Moreover, confocal observations demonstrated that *Td*RL1 is localized within the microtubular spindle during mitosis and exhibits a cytoplasmic distribution with a spotty pattern near the nuclear periphery in interphase cells indicative of its putative role in mitosis and in the microtubular machinery [26]. In contrast, the *Td*PP1-*At*-I2 complex was localized in the cytoplasm and the nucleus suggesting that the two PP1-PIP complexes may have divergent cellular functions or drive the phosphatase to different cellular compartments. Our assumption is reinforced by the fact that, across kingdoms, the subcellular localization in the nucleus, cytoplasm, or microtubules of PP1c is intricately controlled by the interactions with its regulatory subunits, hence allowing its participation in multiple signaling pathways in a highly controlled manner. For example, PP1c has been shown to be essential for the cytoskeleton in neurons through interactions with scaffold proteins like spinophilin, regulating microtubule dynamics [42], and epidermal cells of *topp4-1* mutants present an altered organization of cortical microtubules and F-actin [43].

The specific interaction between *Td*PP1 and *At*-I2 is consistent with the known role of I2 as a “classical” and conserved inhibitor of PP1s across species [21]. The present data show that *Td*RL1 is like I2, acting as a PIP inhibitor, particularly at high concentrations. These inhibitory effects are likely driven by the conserved structural features of PIPs, which include key motifs that facilitate PP1 binding and regulation. The RVxF motif, found in most PIPs, anchors them to PP1’s hydrophobic groove, ensuring high-affinity interaction without necessarily participating in the inhibition [1,19,44]. In addition to the RVxF motif, poly-lysine-rich regions, through electrostatic interactions with PP1’s acidic grooves, contribute to the strength and specificity of the interaction [44] while proline-rich regions provide flexibility, allowing PIPs to form dynamic interactions with PP1, which is essential for fine-tuning their regulatory functions [1]. Similarly, glutamine- or glutamic-acid-rich regions further enhance the binding specificity by contributing to the acidic charge distribution, thus stabilizing the PP1-PIP complex [1]. The *At*-I2 protein contains one RVxF motif (RVQW, located at positions 8 to 11) and an IDoHA motif at positions 125 to 128 (HYDE). The latter motif helps in blocking the phosphatase active site by binding to hydrophobic and acidic grooves of PP1 [45]. Thanks to these motif arrangements, I2 has been shown to act as a competitive inhibitor but also as a chaperone during PP1 folding, stabilizing the enzyme’s nascent structure [45]. Moreover, the interplay between these motifs and their phosphorylation status allows for dynamic regulation of PP1 [20]. Most importantly, according to Mahjoubi et al. [25] *Td*RL1 contains several conserved motifs crucial for its interactions with PP1s including RVxF motifs (_151_PLLF_154_; _211_KPVEF_215_) as well as regions rich in lysine (at position 127–139, Appendix A), glutamate, and proline (215–220) residues, both surrounding the RVxF motif, similarly to the mammalian PP1 inhibitor MYPT1 and other PIPs [17,18]. However, for RSS1, the rice ortholog of *Td*RL1, there is currently no experimental evidence of its inhibitory activity on OsPP1a phosphatase despite its strong binding to OsPP1a through a minimal region (105–207) comprising two RVxF motifs (_103_KTVIF_107_; _165_HVLF_168_) and an Asp-Glu rich region [23,24]. Furthermore, residues flanking these motifs can also be phosphorylated, modifying their binding capacity [12]. Several putative phosphorylation sites are also present in *Td*RL1, which may also affect its binding to PP1 [25].

Previous studies on *Arabidopsis* TOPP4 have shown that the T247M mutation results in an increased accumulation of phosphorylated DELLAs, which suggests a decrease in phosphatase activity towards this substrate. This mutation does not disrupt the interaction between TOPP4 and DELLAs but rather affects the phosphatase activity of TOPP4 [6]. In the case of *Td*PP1, the T243M mutation induced a moderate decrease in the phosphatase activity (around 30%) compared to the H135A mutation (85%). Both *At*-I2 and *Td*RL1 could inhibit the activity of *Td*PP1-T243M, meaning that their interaction with *Td*PP1 was not lost. Therefore, both T243M mutation in *Td*PP1 and the T247M mutation in TOPP4 seem to diminish the phosphatase activity without compromising the ability of these enzymes to bind their regulators. However, it is worth noting that *Td*RL1 inhibited the phosphatase activity of the T243M mutant form with slightly less efficiency than it inhibited the wild-type enzyme. The replacement of a polar amino acid with the larger methionine residue might weaken the binding strength. It is worth noting that *Td*PP1-T243 and TOPP4-T247 are putatively phosphorylated (Appendix A) and their replacement by methionine might hinder phosphorylation and therefore the activity of these mutant forms.

On another hand, the mutation of the aspartate of the DxVDRG motif in PP1, which stabilizes the protonation of the catalytic histidine residue within the GNHE via a salt bridge, was found to result in a significant reduction in *Td*PP1’s phosphatase activity [46]. Similarly, His to Ala mutation, known to disrupt the phosphatase activity of recombinant PP1γ, resulted in the production of unstable, insoluble, and inactive proteins, underscoring its essential role in maintaining both the activity and structural integrity of the enzyme [46]. While mutations in the PP1 active site, such as the catalytic histidine, can dramatically impact the enzyme’s catalytic activity, they do not necessarily impair PP1’s interactions with regulatory proteins. In the case of the *Td*PP1, H135A mutation slightly impacted its inhibition by the *Td*RL1 regulator without affecting the inhibitory potential of *At*-I2. In animals, each PIP seems to have its specific way to bind PP1 [12] and PIP’s binding code is non-exclusive, meaning that the binding of a PIP does not prevent the binding of a second PIP [1]. One cannot exclude that the weaker inhibition potential of *Td*RL1 compared to *At*-I2 might be due to *Td*RL1’s need of a helper PIP. Alternatively, as several PIPs need to be phosphorylated to bind to PP1s [1], it is possible that in a phosphorylated form, *Td*RL1 could be a more potent inhibitor of the PP1.

Additionally, the structural insights provided by the 2O8A complex of mouse PP1c with inhibitor-2 (I2) [45] are remarkably consistent with our findings on *Td*PP1 and its inhibitor *At*-I2. The well-superimposed phosphatase domain of *Td*PP1, with an RMSD of 0.1 Å, confirms the structural conservation of this domain across species. The ordered regions of I2 in the 2O8A structure, particularly the helix-turn-helix region (residues 90–146) that blocks the active site entry, are mirrored in our results, where the *At*-I2 110–137 fragment interacts similarly with *Td*PP1. The specific interactions between *Td*PP1 and *At*-I2, including ion pairing, hydrogen bonding, and hydrophobic interactions, are also supported by the 2O8A structure.

The comparison between *Td*PP1-*Td*RL1 and *Td*PP1-I2 complexes with the 2O8A structure (the mouse PP1-I2 complex) reveals similarities and differences in the structural and interaction profiles of the enzyme and their inhibitors. Both PP1-I2 complexes show that the I2 inhibitor binds to the active site, with a similar inhibitory mechanism where the C-terminal tails of both mouse and *Arabidopsis* proteins block the enzyme’s active site. However, in the 2O8A (mouse PP1-I2) complex, the C-terminal tail of mouse I2 is disordered whereas in the *Td*PP1-*At*-I2 complex, the C-terminal tail of *At*-I2 appears more ordered and wraps around the *Td*PP1 enzyme, stabilizing the complex, which may influence the binding dynamics and inhibitory potency between the two systems. Despite these structural variations, both complexes exhibit a similar mode of inhibition, where the helix-turn-helix region (residues 90–146 of I2) blocks the active site in both cases.

The interaction between the phosphatase *Td*PP1 and the intrinsically disordered protein *Td*RL1 shares some similarities with the previously described 2O8A structure of the PP1-I2 complex. Both complexes involve direct binding of the disordered protein to the phosphatase’s active site, with the C-terminal peptide of *Td*RL1 and the RVxF motif of mouse I2 interacting with residues at the entry of the catalytic pocket to block substrate access. However, while *Td*RL1 binding induces a more rigid conformation, trapping the phosphatase by closing the N-terminal thread, I2 remains largely disordered upon binding PP1. Additionally, the *Td*PP1-*Td*RL1 complex is stabilized by a network of ionic interactions and a strong hydrophobic patch, which is also observed in the 2O8A structure. However, the 2O8 structure highlights the importance of ionic interactions in stabilizing the PP1-I2 complex, with several salt bridges observed at the interface. These observations corroborate previous data, in mammals, on the diversity of structures adopted by the highly disordered PIPs binding PP1s [12] adding complexity in the fine-tuned regulation of PP1 activity.

## 5. Conclusions

We established a specific and functional interaction between *Td*RL1 and *Td*PP1. Notably, our results indicate that *Td*RL1, like *At*-I2 serves as a negative regulator of *Td*PP1, demonstrating significant inhibitory effects on its phosphatase activity, particularly at higher molar ratios. Furthermore, we show here how T243M mutation and H135 of *Td*PP1 affect its activity and may alter its sensitivity to inhibition by *Td*RL1, indicating that specific residues may be important for the interaction dynamics. Interestingly, the predictive three-dimensional model proposed for the *Td*RL1-*Td*PP1 complex shows that the inhibitor surrounds and entraps the phosphatase with its N-terminal tail.

## Figures and Tables

**Figure 1 biomolecules-15-00631-f001:**
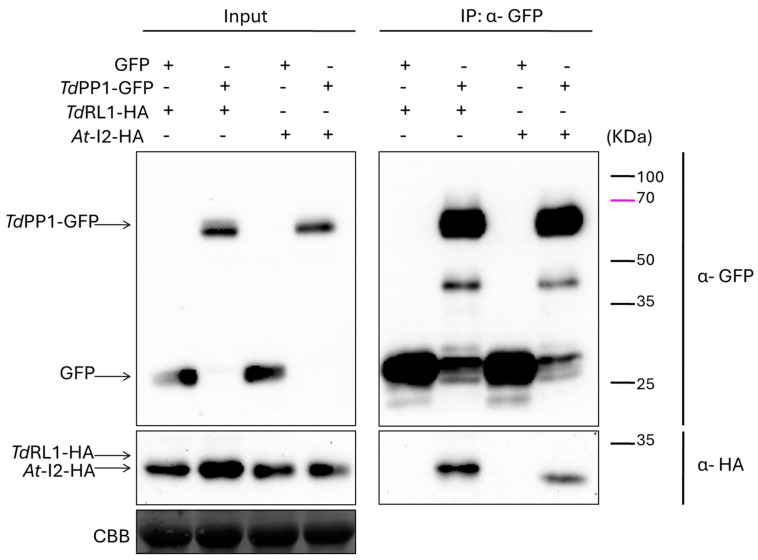
Interaction of *Td*PP1a with *Td*RL1 and *At*-I2 by co-immunoprecipitation (CoIP). Leaves of tobacco were agro-infiltrated with the different combinations: TdPP1-GFP with TdRL1-HA and TdPP1-GFP with *At*-I2-HA. As negative control, the empty vector expressing GFP was combined with either *TdRL1*-HA or *At*-I2-HA. The extracted proteins were divided into input, to check expression by Western blot using anti-GFP for TdPP1 and anti-HA for TdRL1 and *At*-I2, and into IP after immunoprecipitation on anti-GFP beads for Co-IP analysis. SDS page colored with Coomassie Brilliant Blue (CBB) is shown for equal loading. Original images can be found in Appendix A.

**Figure 2 biomolecules-15-00631-f002:**
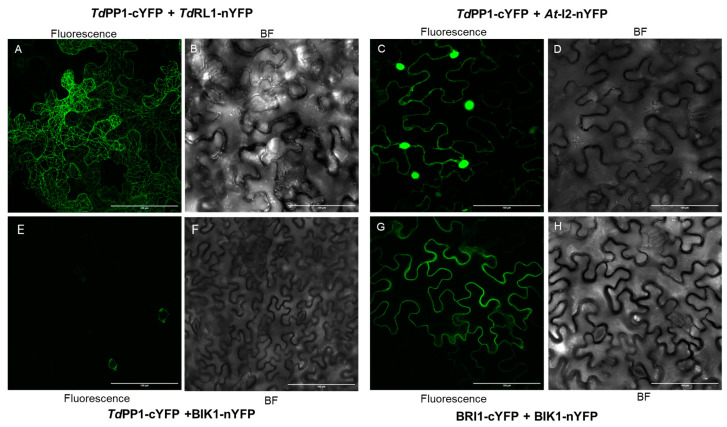
*Td*RL1 interacts with *Td*PP1 in vivo. Bimolecular fluorescence (BiFC) complementation assay in tobacco leaves after co-agro-infiltration of *Td*PP1 and *Td*RL1 (**A**,**B**) or *Td*PP1 and *At*-I2 (**C**,**D**) proteins. *Td*PP1 was fused to the C-terminal half of YFP (PP1-cYFP) and putative interactors were fused to the N-terminal half of YFP. For the positive control, we used BIK1-nYFP and the kinase BRI1-cYFP (**G**,**H**). Negative control PP1-cYFP was tested with BIK1-nYFP (**E**,**F**). Fluorescence and bright field (BF) images are shown for each set of interactions. Scale bar: 100 μm.

**Figure 3 biomolecules-15-00631-f003:**
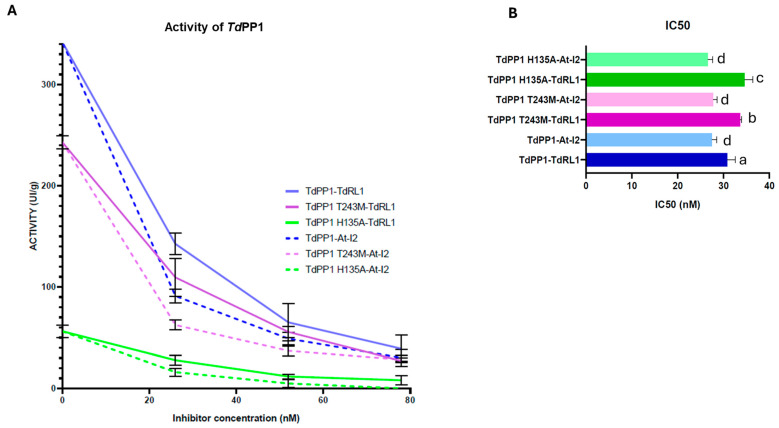
*Td*RL1 inhibits *Td*PP1 phosphatase activity. (**A**) Phosphatase activity of the wild-type and mutant *Td*PP1 forms (T243M and H135A) in the presence of regulators *Td*RL1 and *At*-I2. The phosphatase activity was measured using *p*NPP as a substrate by spectrophotometry at 405 nm in the absence or presence of *Td*RL1 and I2 at three different concentrations. The activity was calculated from three replicates representative of at least three independent experiments. (**B**) IC50 values for *Td*RL1 and *At*-I2 regulators: IC50 represents the concentration of the regulator required to reduce the activity of each *Td*PP1 form by half. The error bars depict the standard deviation (SD) calculated from the mean IC50 values for each combination of *Td*RL1 with the different forms of *Td*PP1 (wild-type, T243M, and H135A) and the same for *At*-I2. Different letters above the bars indicate significant differences between means, as determined by the Tukey HSD test (*p* < 0.05 and *p* < 0.01).

**Figure 4 biomolecules-15-00631-f004:**
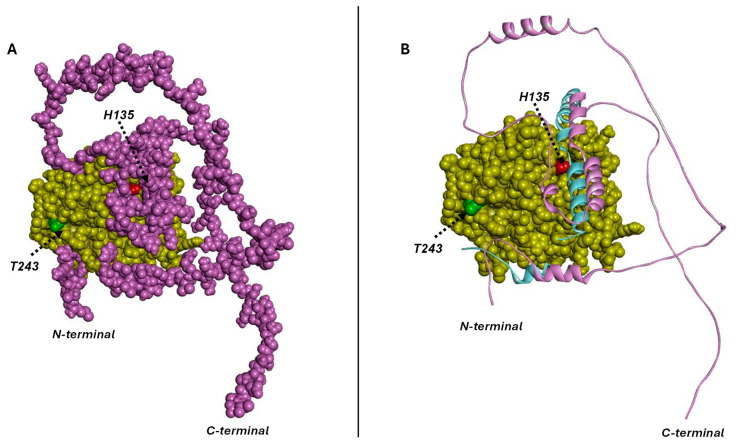
Model structure of the complex between *Td*PP1 (yellow CPK) and its inhibitor I2 (magenta). The structure models were built based on the mouse phosphatase–mouse inhibitor complex (pdb code 2O8A). (**A**) The inhibitor is shown as magenta CPK and the positions of catalytic H135 (in red spheres) and T243 (in green spheres) are indicated. (**B**) The inhibitor is represented as solid ribbon. For the sake of comparison, the structure of the mouse I2 fragment from the template phosphatase inhibitor complex is depicted as solid ribbon (cyan). The *At*-I2 inhibitor interacted with the phosphatase through three main segments, blocking the active site entry. Key interacting residues include R106, H135, S139, R142, I143, Y144, F286, V260, and E262. The H135A and T243M mutants exhibited comparable binding interactions with *At*-I2.

**Figure 5 biomolecules-15-00631-f005:**
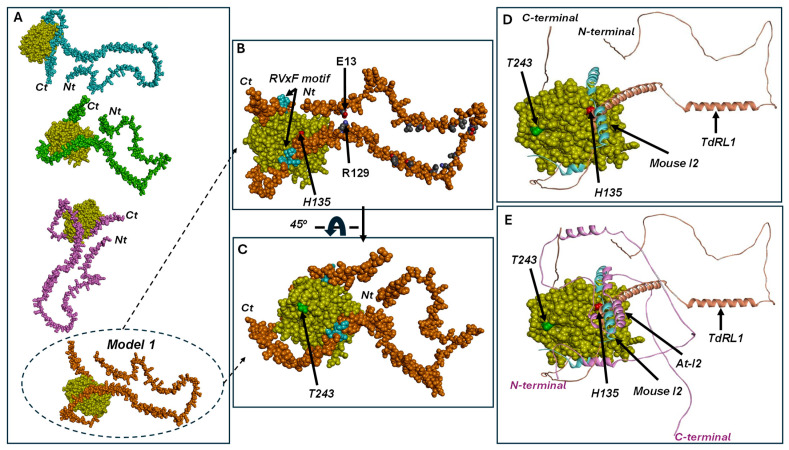
Model structure of the complex between *Td*PP1 (yellow CPK) and its inhibitor *Td*RL1. (**A**) The best complexes generated by Zdock; (**B**–**E**) correspond to model 1 from (**A**). (**B**) CPK model of the complex between the phosphatase and *Td*RL1 showing the catalytic histidine (H135, in red spheres), RVxF motifs of *Td*RL1 (cyan spheres), and interactions stabilizing the *Td*RL1 configuration, including E13-R129 ionic interaction and hydrophobic residues (carbons are in black CPK). The position of the phosphatase catalytic histidine (H135, red spheres) is indicated. (**C**) A 45° view from (**B**) showing the position of the phosphatase residue T243 (green spheres). (**D**) Caption from (**C**) showing the *Td*RL1 inhibitor as solid ribbons (orange) superimposed with the mouse I2–phosphatase complex. Mouse I2 is depicted in cyan solid ribbons. (**E**) Caption from (**D**) superimposed to *At*-I2 complex with the phosphatase (from Figure 4B) (magenta solid ribbons).

## Data Availability

Data are contained within the article and Appendix A.

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
