# Peer review of "The Wheat Intrinsically Disordered Protein TdRL1 Negatively Regulates the Type One Protein Phosphatase TdPP1"

_biomolecules, 2025, doi:10.3390/biom15050631_

Round 1

Reviewer 1 Report

Comments and Suggestions for Authors

Via co-immunoprecipitation, Amor et al. proved that the intrinsically disordered protein RL1 from Triticum durum forms a complex with the type 1 protein phosphatase, PP1. Using bimolecular fluorescence complementation, the authors verified the PP1/RL1 complex and showed that it colocalises with the plant’s cortical microtubular network. Moreover, they verified that RL1 has an inhibitory effect on PP1 and mutated variants thereof. Ultimately, they provided in-silico-generated structural models suggesting the mode of interaction between RL1 and PP1.

While co-immunoprecipitation experiments and bimolecular fluorescence complementation are convincing and sound, the kinetic experiments presented are too preliminary. Characterising the inhibitory potential of RL1 on PP1 activity requires the recording of proper progress curves in the presence of different inhibitor concentrations, ultimately enabling one to calculate Ki values. Instead, the authors simply measured one value after 30 min incubation. This allows one to recognise an inhibitory effect but provides no basis at all for the quantification of inhibition or for the reasonable comparison of the impact of mutations in PP1 on inhibition. Given the fact that the PP1/RL1 complex structures are merely calculated and lack any experimental basis they are utterly overinterpreted in the much too long Discussion. If no attempts were made to determine a cocrystal structure, at least mutations could have been introduced into RL1 to corroborate the speculations made by the authors.

Mistakes and typos:

Throughout the manuscript: in TdPP1, TdRL2, At-I2 and so on, the Td or At should be written in italic since these are short cuts of species names.

Line 51: Change “ABA” to “abscisic acid (ABA)”.

Line 72: Change “… consists in a …” to “… consists of a …”.

Line 87: The statement “… that interacts with hydrophobic residues (R96, H125, Y134, R221, Y272) …” makes absolutely no sense, since the residues in parentheses are not hydrophobic.

Line 112: Change “Nicotiana Benthamiana” to “Nicotiana benthamiana” (here: not italic since the rest of the line is italic).

Lines 148 and 149: Change “minutes” to “min” and seconds” to “s” (as done elsewhere in the manuscript).

Line 192: Change “Fluorescent” to “Fluorescence”.

Line 193: Change “N. Benthamiana” to “N. benthamiana”.

Line 277: See line 193.

Line 285: Change “AtI-2” to “At-I2”.

Line 332: Change “Methionine” to “methionine”.

Line 337: Change “Alanine” to “alanine”.

Figure 3: A and B are interchanged (either in the Figure or in the Legend).

Line 360: Change “… of the of wild-type and mutant …” to “… of the wild-type and mutant …”.

Lines 419 and 420: Change “… catalytic Histidine (H135 in red spheres) … “ to “… catalytic H135 (in red spheres) … “.

Line 502: Change “… inhibition and that …” to “… inhibition that …”.

Line 571: Change “Aspartate” to “aspartate”.

Line 577: Change “Histidine” to “histidine”.

Line 593: Change “helix turn-helix” to “helix-turn-helix”.

Line 602: “Arabidopsis” in italic.

Author Response

Comment 1: While co-immunoprecipitation experiments and bimolecular fluorescence complementation are convincing and sound, the kinetic experiments presented are too preliminary. Characterising the inhibitory potential of RL1 on PP1 activity requires the recording of proper progress curves in the presence of different inhibitor concentrations, ultimately enabling one to calculate Ki values. Instead, the authors simply measured one value after 30 min incubation. This allows one to recognise an inhibitory effect but provides no basis at all for the quantification of inhibition or for the reasonable comparison of the impact of mutations in PP1 on inhibition.

We agree with the reviewer that recording the enzymatic activity in the presence of various concentrations is the best way to characterize the inhibitory power of the ligand. The inhibitory curve is now redrawn, taking into consideration this comment. Moreover, our conclusions about the inhibition potential have been moderated like in the conclusion section (line 606-608: Furthermore, we show here how T243M mutation and H135 of TdPP1 affect its activity and may alter its sensitivity to inhibition by TdRL1, indicating that specific residues may be important for the interaction dynamics).  Accordingly, the caption of figure 3 and phosphatase assays part in the material and methods section have been amended.

Moreover, our assays are based on previous reports on PP1 inhibition potential which use IC50 to determine the inhibition strength (Templeton et al, 2011; Lambrecht  et al, 2013 and other references in the ‘Phosphatase modulators’ book, Ed. Millan, Humana Press, 2013 ISBN: 978-1-62703-561-3).

Comment 2: Given the fact that the PP1/RL1 complex structures are merely calculated and lack any experimental basis they are utterly overinterpreted in the much too long Discussion.

We agree with the reviewer that the model calculated is based on structural prediction of the inhibitors RL1 and I2 which are unstructured proteins. Nevertheless, the complex structure generated by Z dock was based on experimental data which are amino acids of the animals’ I2 analog interacting with the phosphatase as shown by experimental crystallographic data. To avoid overinterpretation, some statements relative to the interactions of the N- or C-terminal ends of the inhibitors (highlighted in yellow) were modified and the potential interacting residues on the enzyme’s surface were deleted from the text, as these peptide ends can be highly flexible. Only interactions close to the enzyme’s active site were maintained in the text.

Comment 3: If no attempts were made to determine a cocrystal structure, at least mutations could have been introduced into RL1 to corroborate the speculations made by the authors.

Thank you for your valuable feedback. We agree that determining a cocrystal structure or introducing mutations into RL1 would significantly strengthen our findings. However, due to time constraints and the lack of appropriate infrastructure, we were unable to perform these additional experiments within the timeframe of the revision. We acknowledge the importance of these suggestions and hope to incorporate them into future studies.

Further comments: Mistakes and typos: Throughout the manuscript: in TdPP1, TdRL2, At-I2 and so on, the Td or At should be written in italic since these are short cuts of species names.

Thanks for pointing out these typos and nomenclature errors. They have all been corrected and are highlighted in yellow in the revised manuscript.

Reviewer 2 Report

Comments and Suggestions for Authors

In the manuscript, the authors performed Co-IP and BiFC to demonstrate the interaction between TdPP1 and TdRL1, tested the inhibitory role TdRL1 to TdPP1, depicted the prediction structure of TdPP1-TdRL1 and described the interaction between TdPP1 and TdRL1. This study contains some interesting findings such as the different subcellular localization between TdRL1 and AtI2 associated with TdPP1, and are valuable for the understanding of regulation mechanism of PP1 in wheat. The results were clearly presented and minor revision need to be done before this manuscript could be accepted for publication.

Minor comments:

(1) For BiFC experiments, BRI1 is a membrane protein and BIK1 is a membrane associated protein, while TdPP1, TdRL1 and AtI2 are soluble proteins based on previous results and this manuscript. For negative control, it's better to choose soluble protein or proteins in the same subcellular compartments. The lack of fluorescent could be explained as the separate subcellular localization of TdPP1 and BRI1 or BIK1. If there's no candidate at the moment, the nYFP  or cYFP alone may also serve as the negative control. If there's no time for any experiment, switch the PP1-cYFP+BIK-nYFP from Figure S1 with Figure 2GH, and emphasize the possible cytosol localization in vivo, though it is still not an ideal control target.

(2) Does AlphaFold3 generate similar results for TdPP1 TdRL1 complex? What's the advantage of Zdock over AlphaFold3? Probably due to the low confident prediction of TdRL1 disordered regions, there might be no improvement or worse prediction. Though structural prediction is valuable, the experimental data is still needed for the real interaction surface identification. For this interaction prediction, what's the rationale for the 1 to 1 ratio for TdPP1 and TdRL1 complex prediction, is there previous reports support this ratio? Or this is the best fit during the prediction trials? Besides the full length, would the conserved fragments be better for complex structure prediction and interaction surface display?

(3) Some typos:

Line75, Aspartate and Histidine, lower case for the first letter.

Supplemental F3. Arabidopsis thaliana should be italic.

Figure S4 '5according', it seems that there's an extra '5'.

Author Response

Comment 1: For BiFC experiments, BRI1 is a membrane protein and BIK1 is a membrane associated protein, while TdPP1, TdRL1 and AtI2 are soluble proteins based on previous results and this manuscript. For negative control, it's better to choose soluble protein or proteins in the same subcellular compartments. The lack of fluorescent could be explained as the separate subcellular localization of TdPP1 and BRI1 or BIK1. If there's no candidate at the moment, the nYFP  or cYFP alone may also serve as the negative control. If there's no time for any experiment, switch the PP1-cYFP+BIK-nYFP from Figure S1 with Figure 2GH, and emphasize the possible cytosol localization in vivo, though it is still not an ideal control target.

We fully agree that the interacting partners used as a negative control should be, located in the same subcellular compartment. However, in previous reports (Amorim-Silva et al, 2019, Plant Cell; https://doi.org/10.1105/tpc.19.00150), soluble proteins have been shown to interact with membrane-associated proteins or nuclear proteins. Additionally, the limited time available for this revision prevented us from conducting further control experiments. Therefore as you suggested, we have swapped the panel shown in Figure S1 (PP1-cYFP+BIK-nYFP) with those shown in the previous version of Figure 2GH.

Comment 2: Does AlphaFold3 generate similar results for TdPP1 TdRL1 complex? What's the advantage of Zdock over AlphaFold3? Probably due to the low confident prediction of TdRL1 disordered regions, there might be no improvement or worse prediction. Though structural prediction is valuable, the experimental data is still needed for the real interaction surface identification.

We appreciate your insightful comment. AlphaFold3 can generate protein complexes as a blind docking method. The complex structure generated by Z dock was based on experimental data which are amino acids of the animal’s I2 analog interacting with the phosphatase and residues from the enzyme’s active site identified in the crystallographic experimental data. Given the high amino acid identity between the animal’s and plant’s phosphatases, a similar binding mode with the regulator is highly expected. Even if the I2 family is a highly heterogenous group of unstructured proteins, it seems however that binding to the regulated phosphatase enhanced the inhibitor organization which was observed in this work. The blind docking can generate functionally irrelevant complexes. That is the reason why we applied a guided protein-ligand binding strategy by using the enzyme’s active site residues and amino acids from I2 analogous to the animals’ counterpart involved in the complex binding as shown by the experimental crystallographic structure. As no structural information is available for RL1 binding mode, docking was guided only by experimental data regarding active site residues interacting with the inhibitor I2. Interestingly, the binding mode was like that of I2, suggesting a common inhibitory strategy.

Comment 2: For this interaction prediction, what's the rationale for the 1 to 1 ratio for TdPP1 and TdRL1 complex prediction, is there previous reports support this ratio? Or this is the best fit during the prediction trials? Besides the full length, would the conserved fragments be better for complex structure prediction and interaction surface display?

We thank the reviewer for this comment. The regulator-target binding in cell signalling for animals or plants are mostly 1:1 complex, likely due to the large inhibitory protein size. To the best of our knowledge, there is no available evidence on a multimer inhibitory systems for phosphatases,. As reported in the literature reviews, the 1:1 ratio is the basic rapid and efficient protein inhibitory system for signalling proteins. The exception to this rule can be enzymes displaying repetitive domains or modules, which is not the case for this phosphatase.

The conserved inhibitor fragments are representing a small part of the whole inhibitor suggesting the remaining parts would help in binding to the protein target. Furthermore, it seems clear from this work and previous reports that the conserved fragments are important for the active site blockage. This suggests that the remaining inhibitor parts would play a crucial role in binding specificity to the target protein. Similar inhibitory mechanisms are reported for signalling proteins’ regulation by other proteins. 

Other comments:  Some typos:

Line75, Aspartate and Histidine, lower case for the first letter.

Supplemental F3. Arabidopsis thaliana should be italic.

Figure S4 '5according', it seems that there's an extra '5'.

Thanks for pointing out these typos. They have all been corrected and are highlighted in yellow in the revised manuscript.

Round 2

Reviewer 1 Report

Comments and Suggestions for Authors

Although the determination of Ki values would have greatly improved the significance of the manuscript, the present kinetics clearly show that TdRL1 is a strong inhibitor of TdPP1. Thus, the manuscript may be published as it is.